# Cloning, Expression and Evaluation of Thioredoxin Peroxidase-1 Antigen for the Serological Diagnosis of *Schistosoma mekongi* Human Infection

**DOI:** 10.3390/diagnostics12123077

**Published:** 2022-12-07

**Authors:** Atcharaphan Wanlop, Jose Ma. M. Angeles, Adrian Miki C. Macalanda, Masashi Kirinoki, Yuma Ohari, Aya Yajima, Junya Yamagishi, Kevin Austin L. Ona, Shin-ichiro Kawazu

**Affiliations:** 1National Research Center for Protozoan Diseases, Obihiro University of Agriculture and Veterinary Medicine, Obihiro 080-8555, Japan; 2Department of Parasitology, College of Public Health, University of the Philippines Manila, Manila 1000, Philippines; 3Department of Immunopathology and Microbiology, College of Veterinary Medicine and Biomedical Sciences, Cavite State University, Indang 4122, Philippines; 4Department of Tropical Medicine and Parasitology, Dokkyo Medical University, Tochigi 321-0293, Japan; 5Department of Veterinary Medicine, Rakuno Gakuen University, Ebetsu 069-8501, Japan; 6Would Health Organization Regional Office for Southeast Asia, New Delhi 110011, India; 7International Institute for Zoonosis Control, Hokkaido University, Sapporo 001-0020, Japan; 8College of Medicine, University of the Philippines Manila, Manila 1000, Philippines

**Keywords:** *Schistosoma mekongi*, *Schistosoma japonicum*, recombinant antigen, ELISA

## Abstract

*Schistosoma mekongi*, a blood fluke that causes Asian zoonotic schistosomiasis, is distributed in communities along the Mekong River in Cambodia and Lao People’s Democratic Republic. Decades of employing numerous control measures including mass drug administration using praziquantel have resulted in a decline in the prevalence of schistosomiasis mekongi. This, however, led to a decrease in sensitivity of Kato–Katz stool microscopy considered as the gold standard in diagnosis. In order to develop a serological assay with high sensitivity and specificity which can replace Kato–Katz, recombinant *S. mekongi* thioredoxin peroxidase-1 protein (rSmekTPx-1) was expressed and produced. Diagnostic performance of the rSmekTPx-1 antigen through ELISA for detecting human schistosomiasis was compared with that of recombinant protein of *S. japonicum* TPx-1 (rSjTPx-1) using serum samples collected from endemic foci in Cambodia. The sensitivity and specificity of rSmekTPx-1 in ELISA were 89.3% and 93.3%, respectively, while those of rSjTPx-1 were 71.4% and 66.7%, respectively. In addition, a higher Kappa value of 0.82 calculated between rSmekTPx-1 antigen ELISA and Kato–Katz confirmed better agreement than between rSjTPx-1 antigen ELISA and Kato–Katz (Kappa value 0.38). These results suggest that ELISA with rSmekTPx-1 antigen can be a potential diagnostic method for detecting active human *S. mekongi* infection.

## 1. Introduction

Schistosomiasis, also known as bilharzia, is a parasitic disease caused by blood-dwelling flukeworms or flatworms, belonging to the class Trematoda and genus *Schistosoma* [1]. This remains a serious public health concern in humans worldwide, with more than 230 million people infected with *Schistosoma* spp. and 800 million individuals at risk of infection [2,3]. Human schistosomiasis is mainly caused by *S. mansoni*, *S. haematobium*, *S. intercalatum*, *S. japonicum* and *S. mekongi* [4]. *S. japonicum* and *S. mekongi* are the two species causing Asian zoonotic schistosomiasis. *S. japonicum* is endemic in the Philippines, the People’s Republic of China and parts of Indonesia [5,6]. On the other hand, *S. mekongi* is found in Cambodia and Lao People’s Democratic Republic (Lao PDR) [1]. It has been reported that these two species are closely related based on morphological and molecular studies [7].

Schistosomiasis mekongi was discovered in Lao PDR and Cambodia fifty years ago, and until now, the disease is still a public health concern for residents in the endemic area [7,8,9]. The diagnosis of *S. mekongi* infection is based on the simple and low-cost Kato–Katz technique which detects schistosome eggs in stool samples and has been recommended as the “gold standard” by the World Health Organization (WHO) [10]. The Kato–Katz technique shows high sensitivity in community settings having high schistosomiasis prevalence and it can estimate the intensity of the infection as eggs per gram (EPG) [11]. However, this technique shows decreased sensitivity when it is used in endemic areas with low prevalence, such as the foci in Cambodia and Lao PDR [12]. As such, this may lead to underestimation of the actual prevalence [10,13,14]. In addition, the Kato–Katz technique requires expertise on the egg’s morphological identification under microscopic observation. This technique is also not suitable for large-scale surveillance since it is laborious and time-consuming [15,16]. Therefore, an appropriate diagnostic method that can replace the Kato–Katz technique is needed for effective treatment and accurate prevalence surveys of *S. mekongi* infection in Lao PDR and Cambodia.

It has been reported that ELISA with recombinant antigen showed higher sensitivity than that of the Kato–Katz technique for detecting human schistosomiasis [17,18,19]. Several recombinant antigens have been evaluated for detecting *S. japonicum* infection in humans and animals in the locations where the disease has been nearly eliminated [14,17,20,21,22].

Thioredoxin peroxidase-1, which is expressed in the tegument and as the excretory/secretory products of adult worms and larvae of *S. japonicum* (SjTPx-1), has been identified as a good antigen in ELISA for detecting human [23,24] and animal schistosomiasis [22,25]. In this study, the cDNA coding for TPx-1 of *S. mekongi* was cloned and the recombinant protein was expressed. The recombinant protein was evaluated for its diagnostic performance as an antigen in ELISA for detecting *S. mekongi* infection in humans.

## 2. Materials and Methods

### 2.1. Human Serum Samples

Archived serum samples collected from schistosomiasis mekongi patients and individuals in the disease non-endemic area in Cambodia by National Center for Parasitology, Entomology and Malaria Control, Ministry of Health in Cambodia, were used. Twenty-eight serum samples were collected from endemic areas in Kratie province, with patients whose infections were confirmed by the Kato–Katz technique. Thirty endemic negative serum samples were collected from individuals in Phnom Penn [26]. Thirty non-endemic negative serum samples from healthy United States volunteers (BioreclamationIVT, Baltimore, MD, USA) were used to calculate cut-off values as the mean +3 standard deviation (SD). In addition, 53 serum samples from patients infected with other parasites were also used to evaluate cross-reaction with the recombinant antigens in ELISA, including 48 serum samples from Thai patients infected with *Opisthorchis viverrini* and 5 serum samples from Japanese patients infected with *Paragonimus westermani*.

### 2.2. TPx-1 Sequence

So far, the whole genome information of *S. mekongi* has not yet been available. The sequence information of the gene coding for TPx-1 of *S. mekongi* (*SmekTPx-1*) was retrieved from the *S. mekongi* genome project which is currently underway in collaboration with the Research Center for Zoonosis Control (CZC), Hokkaido University and Veterinary School of Rakuno Gakuen University, Japan (unpublished data). The coding sequence of TPx-1 of *S. japonicum* (*SjTPx-1*) was obtained from GenBank accession no. AB126036.2).

### 2.3. Recombinant Protein Preparation

The recombinant protein of SjTPx-1 was expressed as previously described in [23]. For expression of the recombinant protein of SmekTPx-1, total RNA was isolated from 10 pairs of *S. mekongi* adult worms [27] using TRIzol reagent (Invitrogen, San Diego, CA, USA) following the manufacturer’s protocol. The cDNA was synthesized using the Ready-To-Go T-Primed First Strand Kit (Amersham Biosciences, Chalfont St Giles, UK) with oligo (dT) primer. The double-stranded full-length coding sequence was amplified by PCR. The Primers were designed according to *SmTPx-1* with *Bam*HI and *Xho*I restriction enzyme sites (underlined) as follows: the forward (5′-GC GGA TCC ATG GTA CTT CC-3′) and reverse primers (5′-GC CTC GAG TTA GTG ATT AGT TTT AAT TC-3′), respectively. The PCR product was cloned into pCR 2.1-TOPO vector (Invitrogen, Carlsbad, CA, USA). The identity of the cloned sequence with the sequence in the draft genome was confirmed by sequencing using an ABI Prism 3100 Genetic Analyzer (Applied Biosystems, Carlsbad, CA, USA). The coding sequence was inserted into pET28 vector (Novagen, Madison, WI, USA) and the plasmid was transformed into *Escherichia coli* Rosetta (DE3) (Novagen) and was grown in LB medium (Sigma-Aldrich, St. Louis, MO, USA) supplemented with 50 µg/mL of kanamycin. Induction of the expression of the recombinant proteins with 0.5 mM isopropyl-thio-β-D-galactoside (IPTG) was done in the SOB medium (BD, Sparks, MD, USA) and then maintained at 37 °C for 3 h. Using nickel nitrilotriacetic acid (Ni-NTA) agarose (Qiagen, Hilden, Germany), the recombinant protein was recovered and purified, followed by elution and dialysis with 20 mM Tris, pH 8.0, according to the manufacturer’s instructions. The quality of the protein was then evaluated using 12% polyacrylamide gel electrophoresis (SDS-PAGE) while the quantity was measured using a bicinchoninic acid (BCA) assay kit (Thermo Fisher Scientific Inc., Rockford, IL, USA). The recombinant proteins were stored in aliquots at −80 °C until used.

### 2.4. Enzyme-Linked Immunosorbent Assay (ELISA)

Indirect ELISA was performed as described in our previous studies [19,23]. In brief, a 96-well polystyrene plate (Thermo Fisher Scientific) was coated with 200 ng in 100 μL/well of recombinant antigens diluted with carbonate/bicarbonate buffer (pH 9.6) at 4 °C overnight. Each well was washed three times with phosphate buffered saline (PBS) containing 0.05% Tween 20 (T-PBS) and blocking was done for 20 min at room temperature with 130 μL/well of T-PBS containing 1% bovine serum albumin. One hundred μL of serum sample diluted at 1:400 in blocking buffer was added into each well in triplicate and incubated at 37 °C for 1 h. After washing the wells with T-PBS for three times, 100 μL of horseradish peroxidase-conjugated anti-human IgG goat serum (Proteintech Group, Inc., Rosemont, IL, USA) as the secondary antibody which had been diluted at 1:10,000 with blocking buffer was added into each well. The plate was incubated at 37 °C for 1 h. After washing the wells with T-PBS for three times, a color reaction was induced by adding the Peroxidase Substrate Solution (KPL, Gaithersburg, MD, USA) and the optical density (OD) was measured at 450 nm using Multiskan SkyHigh Microplate Spectrophotometer (Thermo Fisher Scientific).

### 2.5. Statistical Analysis

The sensitivity, specificity, positive predictive value (PPV) and negative predictive value (NPV) for the recombinant antigens were calculated using the online software MedCalc at https://www.medcalc.org/calc/diagnostic_test.php (accessed on 12 August 2022). An agreement between the ELISA and the Kato–Katz technique for detecting the patients was estimated as a Kappa value [28].

## 3. Results

### 3.1. Cloning and Sequencing of SmekTPx-1

The coding sequence of SmekTPx-1 was successfully amplified and sequenced. The 555 bp of the sequence coded a protein comprising 184 amino acids. SmekTPx-1 showed 94.8% identity with the coding sequence of SjTPx-1 (GenBank accession no. AB126036.2). SmekTPx-1 showed 81.8% identity with the coding sequence of *S. mansoni* TPx-1 (GenBank accession no. AF121199.1) (Appendix A).

### 3.2. Expression and Purification of Recombinant Antigens

The coding sequence of recombinant antigen SmekTPx-1 was successfully expressed as a 6xHis-tag fusion protein and was assessed by SDS-PAGE. The SDS-PAGE and Western blotting experiments confirmed rSmekTPx-1 with the expected molecular weight of 22 kDa including the 6x-His tag (Appendix A, respectively).

### 3.3. ELISA

ELISA results showed that rSmekTPx-1 antigen detected 25 out of 28 patients’ serum samples as positive and confirmed 20 out of 30 serum samples from individuals in the disease non-endemic area as negative. On the other hand, rSjTPx-1 antigen detected 20 out of 28 patient’s serum samples as positive (Figure 1). In comparison with rSjTPx-1 antigen, rSmekTPx-1 antigen showed higher sensitivity, specificity PPV, and NPV (89.3% vs. 71.4%, 97.3% vs. 66.7%, 92.6% vs. 66.7% and 90.3% vs. 71.4%, respectively) as shown in Table 1.

The rSmekTPx-1 antigen showed cross-reactions with 18 out of 48 samples positive for *O. viverrini* infection, while the rSjTPx-1 antigen showed cross-reactions with 22 samples (Figure 2). Both recombinant antigens did not show cross-reaction with samples positive for *P. westermani* infection (Figure 2). The Kappa value calculated between ELISA with rSmekTPx-1 antigen and Kato–Katz was 0.82, which was higher than that calculated between ELISA with rSjTPx-1 antigen and Kato–Katz (0.38) (Table 1). The results confirmed a good agreement between ELISA with rSmekTPx-1 antigen and Kato–Katz for detecting the patients with *S. mekongi* infection. The results suggested that the ELISA may also detect active infection.

## 4. Discussion

Several recombinant antigens have been evaluated for their diagnostic potential for human and animal schistosomiasis [20,23,29]. Recombinant antigens of the proteins expressed on the tegument of *S. mansoni*, such as rSM200, had been successfully applied for serological diagnosis of the schistosomiasis in humans with high sensitivity and specificity [30]. Another recombinant antigen, rRP26, had also been used in ELISA for detecting *S. mansoni* infection in humans with low cross-reactivity against other parasitic infections [31]. Several recombinant antigens have been evaluated for their potential in detecting acute and chronic *S. japonicum* infections [21]. In contrast, very few studies have been reported so far on serological diagnosis of *S. mekongi* infection with recombinant antigens [32,33]. An immunochromatographic test or ICT has been utilized for *S. mekongi* infection with 78.6% sensitivity and 97.6% specificity [33]. However, cross-reactivity, especially in the setting of co-infection with other helminthic parasites, should be addressed by using other diagnostic tools [34]. The point of care circulating cathodic antigen test (POC-CCA) is one of the sensitive diagnostic methods for detecting circulating antigens of schistosomes including *S. mekongi* that have been evaluated, with higher sensitivity compared to the Kato–Katz technique [35]. Nonetheless, high risk of false positive results from the CCA shared epitopes with several human components, and cross-reactivity between the POC-CCA and opisthorchiasis, should be concerns [36,37]. Sangfuang et al. (2016) evaluated the recombinant antigen of *S. mekongi* cathepsin B (rSmekcatB) for detecting the infection in mice. The rSmekcatB showed 91.7% sensitivity and 100% specificity against the mice infection; however, the antigen was not evaluated for its potential against *S. mekongi* infections in humans [38].

TPxs are members of the peroxidase family, important for antioxidant defense in schistosome parasites [39]. TPxs protect parasites from oxidative stress caused by a variety of peroxides and alkyl hydroperoxides in cells [40]. Four types of TPxs have been successfully cloned and characterized in *S. japonicum,* including SjTPx-1, SjTPx-2, SjTPx-3 and SjPrx-4 [19,41]. Three TPxs, including SmTPx-1, SmTPx-2 and SmTPx-3, have also been cloned and characterized in *S. mansoni* [39,42]. Among TPxs of the parasites, TPx-1 of *S. japonicum* has been reported to be a potential antigen with high sensitivity and specificity in ELISA [20,22,24]. TPx-1 is expressed mainly in the tegument of adult and larval worms of the parasite, and that direct exposure to the host immune system makes the protein a good antigen for serological diagnosis [43]. In the present study, TPx-1 of *S. mekongi* was cloned and the recombinant protein was expressed for evaluation of its diagnostic potential as an antigen in ELISA. The amino acid sequence of *S. mekongi* TPx-1 showed high identity of 93.5% with *S. japonicum* TPx-1, suggesting a close relation between the two species according to the previous studies [7,44,45,46].

The rSmekTPx-1 antigen showed good diagnostic performance in ELISA with high sensitivity, specificity, PPV, NPV and Kappa value against the Kato–Katz technique (Table 1). Although the ELISA results showed that the rSmekTPx-1 could detect those samples from individuals with active infections, the recombinant protein should also be tested in post-treatment samples to confirm its capability to detect only the present infection. However, rSjTPx-1 has been proven to show negative results in post-treatment samples, which might be the same for rSmekTPx-1 [23]. To date, there have been a few studies on ELISA for the diagnosis of *S. mekongi* infection [47]. The soluble egg antigen of *S. mekongi* (SmekSEA) has been used in ELISA for the diagnosis of patients with 100% sensitivity and specificity [48]. However, SmekSEA showed cross-reactivity against other parasitic infections including ascariasis, echinostomiasis, opisthorchiasis and hookworm infection [48,49]. Nickel et al. (2015) reported that soluble adult worm antigen (SWAP) prepared from *S. mansoni* showed 94.5% sensitivity in ELISA for diagnosis of *S. mekongi* infection in humans [50]. However, the preparation of crude antigens such as SEA and SWAP is problematic due to observed low yields and poor quality control [51]. In addition, wide and high cross-reactions against other trematode infections are of concern when they are used in prevalence surveys [23]. In this study, the rSmekTPx-1 antigen showed no cross-reaction with serum samples from paragonimiasis patients. However, it showed 38% cross-reactions with serum samples from opisthorchiasis patients (Figure 2). It is recommended that the amino acid sequence between SmekTPx-1 and *O. viverrini* TPx-1 should be compared in order to modify rSmekTPx-1 by removing the peptide sequence with high identity between these parasite species which may trigger the cross-reaction.

## 5. Conclusions

In the present study, *SmekTPx-1*, the gene coding for *S. mekongi* TPx-1, was cloned and the recombinant protein, rSmekTPx-1, was successfully expressed. The rSmekTPx-1 antigen showed higher sensitivity and specificity as compared to those of the rSjTPx-1 antigen in ELISA with a panel of serum samples. A good agreement between ELISA with rSmekTPx-1 antigen and Kato–Katz suggested that the ELISA can detect an active infection, which requires treatment with praziquantel. The results suggested that rSmekTPx-1 can be a potential antigen in ELISA for diagnosis of schistosomiasis mekongi in the regions where the disease is still endemic. Field evaluation should be done to determine the performance of the rSmekTPx-1 ELISA in areas with low prevalence in the future study.

## Figures and Tables

**Figure 1 diagnostics-12-03077-f001:**
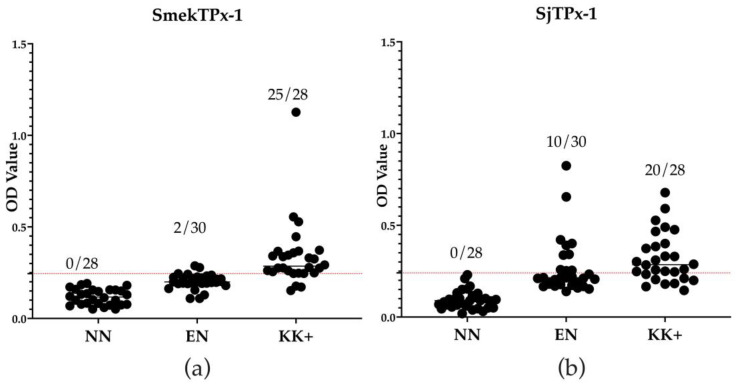
Results of ELISA with rSmekTPx-1 antigen (rSmekTPx-1 ELISA) (**a**) and rSjTPx-1 ELISA (**b**). The panel of serum samples tested included non-endemic negative serum samples from USA volunteers (NN), endemic negative serum samples collected from individuals in Phnom Penh (EN) and serum samples collected from patients in Kratie that were confirmed positive by the Kato–Katz technique (KK+). The cut-off OD values were calculated from the values of 28 NN as mean + 3SD and were presented by the red dotted lines. Numbers in the figures indicate the number of samples with OD value higher than the cut-off values (positive samples)/number of samples tested.

**Figure 2 diagnostics-12-03077-f002:**
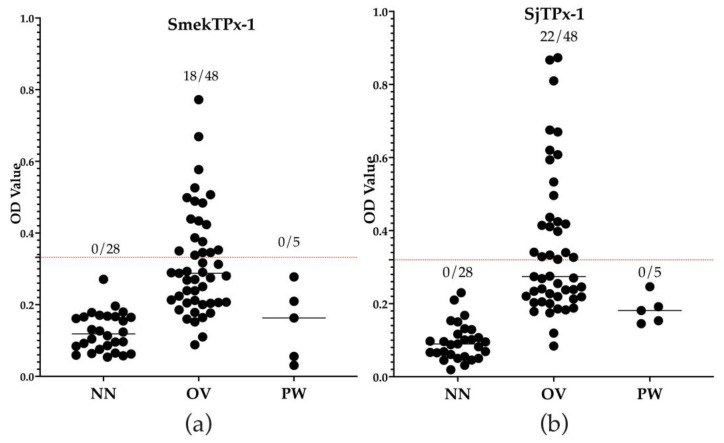
Cross-reactions observed in rSmekTPx-1 ELISA (**a**) and rSjTPx-1 ELISA (**b**) with non-endemic negative serum samples from USA volunteers (NN), serum samples obtained from patients infected with *Opisthorchis viverrini* (OV) and serum samples obtained from patients infected with *Paragonimus westermani* (PW). The cut-off OD values were calculated from the values of 28 NN as mean + 3SD and were presented by the red dotted lines. Numbers in the figures indicate the number of samples with OD value higher than the cut-off values (positive samples)/number of samples tested.

**Table 1 diagnostics-12-03077-t001:** Statistical analysis of ELISA results with rSmekTPx-1 and rSjTPx-1.

Antigen	Sensitivity (%)	Specificity (%)	PPV (%)	NPV (%)	K ^1^
SmekTPx-1	89.3 (95% CI: 71.8–97.3)	93.3 (95% CI: 77.9–99.2)	92.6 (95% CI: 76.5–97.7)	90.3 (95% CI: 76.1–96.5)	0.82
SjTPx-1	71.4 (95% CI: 51.3–86.8)	66.7 (95% CI: 47.2–82.7)	66.7 (95% CI: 53.4–77.7)	71.4 (95% CI: 56.9–82.6)	0.38

^1^: Kappa values were estimated against the Kato–Katz technique.

## Data Availability

The data relating to this manuscript are available upon request.

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
