# Peer review of "Cloning, Expression and Evaluation of Thioredoxin Peroxidase-1 Antigen for the Serological Diagnosis of Schistosoma mekongi Human Infection"

_diagnostics, 2022, doi:10.3390/diagnostics12123077_

Round 1

Reviewer 1 Report

This manuscript is well-written, easy to follow, and contains helpful background information. Other than some minor points mentioned below,

line 157-159 More clarification would benefit the reader. Please explain and identify the number of endemic and non-endemic areas in the method section (Human serum samples).

Author Response

Comment: line 157-159 More clarification would benefit the reader. Please explain and identify the number of endemic and non-endemic areas in the method section (Human serum samples).

Response

We thank the reviewer for the comment. We have revised the manuscript according to the suggestion (lines 77-79). Background information on the samples are now included.

Reviewer 2 Report

Wanlop et al. report on their experience with a new in-house serological assay for the detection of anti-Schistosoma mekongi antibodies based on a newly expressed antigen. The sample size for the assessment is pretty small, however, as little research is done on neglected Schistosoma mekongi infections, even such small works can contribute to a gradual improvement of disease diagnosis and management. Before publication should be considered, I have a few suggestions.

1.) Although I am not a native speaker myself, some typing and Grammar mistakes as well as use of non-idiomatic phrases make the work partly difficult to read. I recommend for a thorough language proof reading either by the authors or by the journal prior to publication.

2.) Methods, sub-heading “Human serum samples”: It is not clear to me how S. mekongi infection (e.g., due to former travelling in endemicity setting) was “definitely” excluded in the “negative control” sample population? Or was it just based on the assumption of “low epidemiological likeliness”? If some of the patients, from whom the “negative control” samples were taken, had indeed been infected, seemingly low specificity values would not necessarily provide hint to non-specific reactions, especially considering the low total sample count.

3.) Methods chapter, subheading “Enzyme-Linked Immunosorbent Assay (ELISA)”: Have any optimization steps been comparatively evaluated during the set-up of the ELISA assay? If not, what was the reason for not doing it? If yes, please provide respective details.

4.) Results chapter, ELISA: It is not clear how exactly the cut-offs were defined. Just to get optimized accuracy values for the paper? What about 95-%-confidence intervals? How reliable will the discrimination be if such intervals are considered for the interpretation of the test results?

5.) Results, line 174: It is not clear to me how the authors come to the conclusion that their serology assay indeed indicates active infection (in comparison to previous, already cured infection)? I have difficulties coming to this conclusion based on the provided data and so, the authors should either weaken this statement or explain more clearly based on which evidence they defend their respective hypothesis.

6.) Discussion, last paragraph: I am not sure if I can really follow the authors very optimistic view on the diagnostic performance of their assay. In populations with low pretest-probability like in Schistosoma eradication settings, the predictive value of a positive result will still be pretty low. This problem is common for helminth serology due to its extensive cross-reactivity and so, I don’t see a reason for not openly addressing this limitation.

Author Response

Thank you very much for your comments and suggestions. They are very valuable and helpful for improving our manuscript. We have revised the manuscript accordingly.

Comment No. 1. Although I am not a native speaker myself, some typing and Grammar mistakes as well as use of non-idiomatic phrases make the work partly difficult to read. I recommend for a thorough language proof reading either by the authors or by the journal prior to publication.

Response: Thank you very much for your comment. The manuscript was checked once again by English speaking co-authors and revised accordingly.

Comment No. 2. Methods, sub-heading “Human serum samples”: It is not clear to me how S. mekongi infection (e.g., due to former travelling in endemicity setting) was “definitely” excluded in the “negative control” sample population? Or was it just based on the assumption of “low epidemiological likeliness”? If some of the patients, from whom the “negative control” samples were taken, had indeed been infected, seemingly low specificity values would not necessarily provide hint to non-specific reactions, especially considering the low total sample count.

Response: Thank you very much for your comment. We thank the reviewer for this comment. Both the negative and positive serum samples for S. mekomgi infection were determined through Kato-Katz stool examination. This was mentioned in the Materials and Methods section (lines 77-79).

Comment No. 3. Methods chapter, subheading “Enzyme-Linked Immunosorbent Assay (ELISA)”: Have any optimization steps been comparatively evaluated during the set-up of the ELISA assay? If not, what was the reason for not doing it? If yes, please provide respective details.

Response: Thank you very much for your comment. We have several publications on the use of recombinant antigens through ELISA for the diagnosis of schistosomiasis japonica in humans and animals. The optimization of the ELISA was done as previously described (Angeles et al., 2011)

Comment No. 4. Results chapter, ELISA: It is not clear how exactly the cut-offs were defined. Just to get optimized accuracy values for the paper? What about 95-%-confidence intervals? How reliable will the discrimination be if such intervals are considered for the interpretation of the test results?

Response: Thank you very much for your comment. The cut-off values were computed using ELISA OD values of 28 non-endemic negative serum samples collected from healthy US volunteers calculated as Mean + 3SD. This was mentioned in the Materials and Methods section (line 80-83). In addition, 95% confidence intervals were demonstrated in Table 1.

Comment No. 5. Results, line 174: It is not clear to me how the authors come to the conclusion that their serology assay indeed indicates active infection (in comparison to previous, already cured infection)? I have difficulties coming to this conclusion based on the provided data and so, the authors should either weaken this statement or explain more clearly based on which evidence they defend their respective hypothesis.

Response: Thank you very much for your comment. The ELISA results for positive and negative of the infection have strong agreement with Kato-Katz stool examination which can detect the patient who excretes the egg and requires treatment. Since we haven’t checked post-treatment samples from previously infected individuals, we could not claim that SmekTPx-1 could only detect active but not the past infections. However, we have previously evaluated SjTPx-1 with samples from individuals one year after praziquantel treatment and showed negative results. This could possibly be true also with SmekTPx-1. We cited our previous publication and explained as “Although the ELISA results showed that the rSmekTPx-1 could detect those samples from individuals with active infection, the recombinant protein should also be tested in post-treatment samples to confirm its capability to detect only the present infection. However, rSjTPx-1 has been proven to show negative results in post-treatment samples which might be the same for rSmekTPx-1[23].” (line 228-232).

Comment No. 6. Discussion, last paragraph: I am not sure if I can really follow the authors very optimistic view on the diagnostic performance of their assay. In populations with low pretest-probability like in Schistosoma eradication settings, the predictive value of a positive result will still be pretty low. This problem is common for helminth serology due to its extensive cross-reactivity and so, I don’t see a reason for not openly addressing this limitation.

Response: Thank you very much for your comment. The use of parasite specific recombinant proteins decreases cross-reactivity results, thus decreasing the likelihood of finding false positive results. In our previous and current studies, we have evaluated our recombinant proteins such as SjTPx-1 for cross-reaction with other parasitic infection including trichuriasis, paragonimiasis, malaria, amoebiasis among others and showed none to very low cross-reactivity. The modification of the recombinant antigen was suggested in the Discussion section (lines 240-247). As you have suggested an additional evaluation should be made for improving the performance of the ELISA system. We therefore added the following sentence in the Conclusion section. “Field evaluation should be done to determine the performance of the rSmekTPx-1 ELISA in areas with low prevalence in the future study.” (lines 256-257).

Reviewer 3 Report

Wanlop and coauthors present original data obtained using recombinant thioredoxin peroxidase-1 (SmTPx-1) as candidate diagnostic antigen for Schistosmiasis mekongi. Results obtained with a moderate number of sera (30 negative, 28 patients, 53 other infections) are promising, yielding 89,3% sensitivity and 93,3% specificity. Results obtained with SjTPx-1 were less satisfactory, demonstrating a better diagnostic result with the enzyme from the cognate species.

Major points

1) The discussion should be a bit more exhaustive and include other  published work on S. mekongi diagnosis, e.g. the lateral flow assay described by Rodpai et al. (PMID: 34981973), and I would expect to find the metaanalysis by Rahman et al. cited (PMID: 33730048). There is not much published but thisis an opportunity to compare owne's own method with those described by other groups. Pleaseperfrom a thorough search of Pubmed and include any relevant paper

2) No data are shown regarding the recombinant expression of the candidate diagnostic antigen. It is important to add data from expression (SDS-PAGE, Western Blotting) to document successful expression and purification. The results of IMAC purification should also be shown.

3) The uploaded manuscript did not include any supplementary data- what does Figure S1 show?

Minor points

Please check/amend the following minor issues:

Using Sm as abbreviation for S. mekongi could lead to confusion with Sm for S. mansoni. Consider using SmeTPx-1 or SmekTPx-1 as alternative

Line 22: have resulted in a decrease in prevalence...

Line 26: through (or via), not thru

Line 37: Schistosomiasis, also well-known as bilharzia. Consider removing 'well'. It is subjective. My guess is that >99.9% of humanity do not know what Bilharzia or Schistosomes are

Line 69: larvae

Line 70/71:  "In this study, cDNA coding for TPx-1" (not the gene: a gene also includes reagulatory elements such as promoters, etc.)

Line 103: primers

Line 108: transformed, not transfected

Lines 110/111 Assuming temperature was 37°C but should be mentioned

Author Response

Major points

1) The discussion should be a bit more exhaustive and include other published work on S. mekongi diagnosis, e.g., the lateral flow assay described by Rodpai et al. (PMID: 34981973), and I would expect to find the meta-analysis by Rahman et al. cited (PMID: 33730048). There is not much published, but this is an opportunity to compare owne's own method with those described by other groups. Please perfrom a thorough search of Pubmed and include any relevant paper

Response: Thank you very much for your suggestions. We have included statements comparing our work with those published methods. We also cited the suggested publications and papers from Pubmed that are relevant to the present study [33-38] (line 199-207).

2) No data are shown regarding the recombinant expression of the candidate diagnostic antigen. It is important to add data from expression (SDS-PAGE, Western Blotting) to document successful expression and purification. The results of IMAC purification should also be shown.

Response: Thank you very much for your comment. The results of SDS-PAGE and Western Blotting were provided as Supplementary Figure S2 and S3, respectively (lines 148-152). In order to support the description in the legend for Figure S2 and 3, a sentence of “according to the manufacturer’s instructions” was added to the Materials and Methods section (lines114-115). However, we have only done protein purification using nickel nitrilotriacetic acid (Ni-NTA) agarose (Qiagen, Hilden, Germany) (lines 112-115), but not IMAC purification.

3) The uploaded manuscript did not include any supplementary data- what does Figure S1 show?

Response: Thank you very much for your comment and we apologize for the mistake. We have uploaded the supplementary Figure S1.

Minor points

Please check/amend the following minor issues:

Using Sm as abbreviation for S. mekongi could lead to confusion with Sm for S. mansoni. Consider using SmeTPx-1 or SmekTPx-1 as alternative

Line 22: have resulted in a decrease in prevalence...

Line 26: through (or via), not thru

Line 37: Schistosomiasis, also well-known as bilharzia. Consider removing 'well'. It is subjective. My guess is that >99.9% of humanity do not know what Bilharzia or Schistosomes are

Line 69: larvae

Line 70/71:  "In this study, cDNA coding for TPx-1" (not the gene: a gene also includes reagulatory elements such as promoters, etc.)

Line 103: primers

Line 108: transformed, not transfected88

Lines 110/111 Assuming temperature was 37°C but should be mentioned

Response:  Thank you very much for your suggestions. They were all checked and amended accordingly.